# Differentially Private Contextual Linear Bandits

**Roshan Shariff**
Department of Computing Science
University of Alberta
Edmonton, Alberta, Canada
`roshan.shariff@ualberta.ca`

**Or Sheffet**
Department of Computing Science
University of Alberta
Edmonton, Alberta, Canada
`osheffet@ualberta.ca`

## Abstract

We study the contextual linear bandit problem, a version of the standard stochastic multi-armed bandit (MAB) problem where a learner sequentially selects actions to maximize a reward which depends also on a user provided per-round *context*. Though the context is chosen arbitrarily or adversarially, the reward is assumed to be a stochastic function of a feature vector that encodes the context and selected action. Our goal is to devise private learners for the contextual linear bandit problem.

We first show that using the standard definition of differential privacy results in linear regret. So instead, we adopt the notion of *joint* differential privacy, where we assume that the action chosen on day $t$ is only revealed to user $t$ and thus needn't be kept private that day, only on following days. We give a general scheme converting the classic linear-UCB algorithm into a joint differentially private algorithm using the tree-based algorithm [10, 18]. We then apply either Gaussian noise or Wishart noise to achieve joint-differentially private algorithms and bound the resulting algorithms' regrets. In addition, we give the first lower bound on the *additional* regret any private algorithms for the MAB problem must incur.

## 1 Introduction

The well-known *stochastic multi-armed bandit* (MAB) is a sequential decision-making task in which a learner repeatedly chooses an action (or arm) and receives a noisy reward. The objective is to maximize cumulative reward by *exploring* the actions to discover optimal ones (having the best expected reward), balanced with *exploiting* them. The *contextual* bandit problem is an extension of the MAB problem, where the learner also receives a *context* in each round, and the expected reward depends on *both* the context and the selected action.

As a motivating example, consider online shopping: the user provides a context (composed of query words, past purchases, etc.), and the website responds with a suggested product and receives a reward if the user buys it. Ignoring the context and modeling the problem as a standard MAB (with an action for each possible product) suffers from the drawback of ignoring the variety of users' preferences; whereas separately learning each user's preferences doesn't allow us to generalize between users. Therefore it is common to model the task as a contextual *linear bandit* problem: Based on the user-given context, each action is mapped to a feature vector; the reward probability is then assumed to depend on the *same* unknown linear function of the feature vector across all users.

The above example motivates the need for privacy in the contextual bandit setting: users' past purchases and search queries are sensitive personal information, yet they strongly predict future purchases. In this work, we give upper and lower bounds for the problem of (joint) *differentially private* contextual linear bandits. Differential privacy is the *de facto* gold standard of privacy-preserving data analysis in both academia and industry, requiring that an algorithm's output have very limited dependency on any single user interaction (one context and reward). However, as we later illustrate, adhering to the standard notion of differential privacy (under event-level continual observation) in the contextual bandit requires us to essentially ignore the context and thus incur linear regret. We

therefore adopt the more relaxed notion of *joint differential privacy* [23] which, intuitively, allows us to present the $t$-th user with products corresponding to her preferences, while guaranteeing that all interactions with all users at times $t' > t$ have very limited dependence on user $t$'s preferences. The guarantee of differential privacy under continuous observation assures us that even if all later users collude in an effort to learn user $t$'s context or preference, they still have very limited advantage over a random guess.

## 1.1 Problem Formulation

**Stochastic Contextual Linear Bandits.** In the classic MAB, in every round $t$ a learner selects an *action* $a_t$ from a fixed set $\mathcal{A}$ and receives a *reward* $y_t$. In the (stationary) *stochastic* MAB, the reward is noisy with a fixed but unknown expectation $\mathbb{E}[y_t \mid a_t]$ that depends only on the selected action. In the stochastic contextual bandit problem, before each round the learner also receives a *context* $c_t \in C$ — the expected reward $\mathbb{E}[y_t \mid c_t, a_t]$ depends on both $c_t$ and $a_t$. It is common to assume that the context affects the reward in a linear way: map every context-action pair to a *feature vector* $\phi(c, a) \in \mathbb{R}^d$ (where $\phi$ is an arbitrary but known function) and assume that $\mathbb{E}[y_t \mid c_t, a_t] = \langle \boldsymbol{\theta}^*, \phi(c_t, a_t) \rangle$. The vector $\boldsymbol{\theta}^* \in \mathbb{R}^d$ is the key unknown parameter of the environment which the learner must discover to maximize reward. Alternatively, we say that on every round the learner is given a *decision set* $\mathcal{D}_t := \{\phi(c_t, a) \mid a \in \mathcal{A}\}$ of all the pre-computed feature vectors: choosing $\boldsymbol{x}_t \in \mathcal{D}_t$ effectively determines the action $a_t \in \mathcal{A}$. Thus, the contextual stochastic linear bandit framework consists of repeated rounds in which the learner: (i) receives a *decision set* $\mathcal{D}_t \subset \mathbb{R}^d$; (ii) chooses an *action* $\boldsymbol{x}_t \in \mathcal{D}_t$; and (iii) receives a stochastic *reward* $y_t = \langle \boldsymbol{\theta}^*, \boldsymbol{x}_t \rangle + \eta_t$. When all the $\mathcal{D}_t$ are identical and consist of the standard basis vectors, the problem reduces to standard MAB.

The learner's objective is to maximize cumulative reward, which is equivalent to minimizing *regret*: the extra reward a learner would have received by always choosing the best available action. In other words, the regret characterizes the cost of having to *learn* the optimal action over just *knowing* it beforehand. For stochastic problems, we are usually interested in a related quantity called *pseudo-regret*, which is the extra *expected* reward that the learner could have earned if it had known $\boldsymbol{\theta}^*$ in advance. In our setting, the cumulative pseudo-regret after $n$ rounds is $\widehat{R}_n := \sum_{t=1}^{n} \max_{\boldsymbol{x} \in \mathcal{D}_t} \langle \boldsymbol{\theta}^*, \boldsymbol{x} - \boldsymbol{x}_t \rangle$.[1]

**Joint Differential Privacy.** As discussed above, the context and reward may be considered private information about the users which we wish to keep private from all *other* users. We thus introduce the notion of jointly differentially private learners under continuous observation, a combination of two definitions [given in 23, 18]. First, we say two sequences $S = \langle (\mathcal{D}_1, y_1), (\mathcal{D}_2, y_2), \ldots, (\mathcal{D}_n, y_n) \rangle$ and $S' = \langle (\mathcal{D}'_1, y'_1), \ldots, (\mathcal{D}'_n, y'_n) \rangle$ are *t-neighbors* if for all $t' \neq t$ it holds that $(\mathcal{D}_{t'}, y_{t'}) = (\mathcal{D}'_{t'}, y'_{t'})$.

**Definition 1.** A randomized algorithm $A$ for the contextual bandit problem is $(\varepsilon, \delta)$-*jointly differentially private* (JDP) under continual observation if for any $t$ and any pair of $t$-neighboring sequences $S$ and $S'$, and any subset $\mathcal{S}_{>t} \subset \mathcal{D}_{t+1} \times \mathcal{D}_{t+2} \times \cdots \times \mathcal{D}_n$ of sequence of actions ranging from day $t + 1$ to the end of the sequence, it holds that $\mathbb{P}(A(S) \in \mathcal{S}_{>t}) \leq e^{\varepsilon} \mathbb{P}(A(S') \in \mathcal{S}'_{>t}) + \delta$.

The standard notion of differential privacy under continual observation would require that changing the context $c_t$ cannot have much effect on the probability of choosing action $a_t$ — even for round $t$ itself (not just for future rounds as with JDP). In our problem formulation, however, changing $c_t$ to $c'_t$ may change the decision set $\mathcal{D}_t$ to a possibly disjoint $\mathcal{D}'_t$, making that notion ill-defined. Therefore, when we discuss the impossibility of regret-minimization under standard differential privacy in Section 5, we revert back to a fixed action set $\mathcal{A}$ with an explicit per-round context $c_t$.

## 1.2 Our Contributions and Paper Organization

In this work, in addition to formulating the definition of JDP under continual observation, we also present a framework for implementing JDP algorithms for the contextual linear bandit problem. Not surprisingly, our framework combines a tree-based privacy algorithm [10, 18] with a linear upper confidence bound (LinUCB) algorithm [13]. For modularity, in Section 3 we analyze a family of linear UCB algorithms that use different regularizers in every round, under the premise that the

regularizers are PSD with bounded singular values. Moreover, we repeat our analysis twice — first we obtain a general $\tilde{O}(\sqrt{n})$ upper bound on regret; then, for problem instances that maintain a $\Delta$ reward gap separating the optimal and sub-optimal actions, we obtain a polylog$(n)/\Delta$ regret upper bound. Our leading application of course is privacy, though one could postulate other reasons where such changing regularizers would be useful (e.g., if parameter estimates turn out to be wrong and have to be updated). We then plug two particular regularizers into our scheme: the first is a privacy-preserving mechanism that uses additive Wishart noise [30] (which is always PSD); the second uses additive Gaussian noise [19] (shifted to make it PSD w.h.p. over all rounds). The main term in the two regret bounds obtained by both algorithms is $\tilde{O}(\sqrt{n} \cdot d^{3/4}/\sqrt{\varepsilon})$ (the bound itself depends on numerous parameters; a notation list appears in Section 2). Details of both techniques appear in Section 4. Experiments with a few variants of our algorithms are detailed in Section D of the supplementary material. In Section 5 we also give a lower bound for the $\varepsilon$-differentially private MAB problem. Whereas all previous work on the private MAB problem uses standard (non-private) bounds, we show that *any* private algorithm must incur *an additional* regret of $\Omega(k \log(n)/\varepsilon)$. While the result resembles the lower bound in the adversarial setting, the proof technique cannot rely on standard packing arguments [e.g. 20] since the input for the problem is stochastic rather than adversarial. Instead, we rely on a recent coupling argument [22] to prove any private algorithm must substantially explore suboptimal arms. By showing that the contextual bandit problem does not become much harder by adding privacy, we open the possibility of machine learning systems that are useful to their users without significantly compromising their privacy.

**Future Directions.**    The linear UCB algorithm we adapt in this work is a canonical approach to the linear bandit problem, using the principle of "optimism in the face of uncertainty." However, recent work [24] shows that all such "optimistic" algorithms are sub-optimal, and instead proposes adapting to the decision set in a particular way by solving an intricate optimization problem. It remains an open question to devise a private version of this algorithm which interpolates between UCB and fine-tuning to the specific action set.

## 1.3   Related Work

**MAB and the Contextual Bandit Problem.**    The MAB dates to the seminal work of Robbins [28], with the UCB approach developed in a series of works [8, 4] culminating in [6]. Stochastic linear bandits were formally first introduced in [3], and [5] was the first paper to consider UCB-style algorithms. An algorithm that is based on a confidence ellipsoid is described by [13], with a variant based on ridge regression given in [12], or explore-then-commit variant in [29], and a variant related to a sparse setting appears in [2]. Abbasi-Yadkori et al. [1] gives an instance dependent bound for linear bandits, which we convert to the contextual setting.

**Differential Privacy.**    Differential privacy, first introduced by Dwork et al. [17, 16], is a rigorous mathematical notion of privacy that requires the probability of any observable output to change very little when any single datum changes. (We omit the formal definition, having already defined JDP.) Among its many elegant traits is the notion of group privacy: should $k$ datums change then the change in the probability of any event is still limited by (roughly) $k$ times the change when a single datum was changed. Differential privacy also composes: the combination of $k$ $(\varepsilon, \delta)$-differentially private algorithms is $\left(O(k\varepsilon^2 + 2\sqrt{k \log(1/\delta')}), k\delta + \delta'\right)$-differentially private for any $\delta' > 0$ [14].

The notion of differential privacy under continual observation was first defined by Dwork et al. [18] using the **tree-based algorithm** [originally appearing in 10]. This algorithm maintains a binary tree whose $n$ leaves correspond to the $n$ entries in the input sequence. Each node in the tree maintains a noisy (privacy-preserving) sum of the input entries in its subtree — the cumulative sums of the inputs can thus be obtained by combining at most $\log(n)$ noisy sums. This algorithm is the key ingredient of a variety of works that deal with privacy in an online setting, including counts [18], online convex optimization [21], and regret minimization in both the adversarial [31, 34] and stochastic [26, 33] settings. We comment that Mishra and Thakurta [26] proposed an algorithm similar to our own for the contextual bandit setting, however (i) without maintaining PSD, (ii) without any analysis, only empirical evidence, and (iii) without presenting lower bounds. A partial utility analysis of this algorithm, in the reward-privacy model (where the context's privacy is not guaranteed), appears in the recent work of Neel and Roth [27]. Further details about achieving differential privacy via additive noise and the tree-based algorithm appear in Section A of the supplementary material. The related problem of private linear regression has also been extensively studied in the offline setting [11, 7].

## 2 Preliminaries and Notation

We use ***bold*** letters to denote vectors and bold ***CAPITALS*** for matrices. Given a $d$-column matrix $\boldsymbol{M}$, its *Gram matrix* is the $(d \times d)$-matrix $\boldsymbol{M}^\mathsf{T}\boldsymbol{M}$. A symmetric matrix $\boldsymbol{M}$ is positive-semidefinite (PSD, denoted $\boldsymbol{M} \succeq 0$) if $\boldsymbol{x}^\mathsf{T}\boldsymbol{M}\boldsymbol{x} \geq 0$ for any vector $\boldsymbol{x}$. Any such $\boldsymbol{M}$ defines a norm on vectors, so we define $\|x\|_M^2 = \boldsymbol{x}^\mathsf{T}\boldsymbol{M}\boldsymbol{x}$. We use $\boldsymbol{M} \succeq \boldsymbol{N}$ to mean $\boldsymbol{M} - \boldsymbol{N} \succeq 0$. The Gaussian distribution $\mathcal{N}(\mu, \sigma^2)$ is defined by the density function $(2\pi\sigma^2)^{-1/2} \exp(-(x-\mu)^2/2\sigma^2)$. The squared $L_2$-norm of a $d$-dimensional vector whose coordinates are drawn i.i.d. from $\mathcal{N}(0,1)$ is given by the $\chi^2(d)$ distribution, which is tightly concentrated around $d$. Given two distributions $\mathbb{P}$ and $\mathbb{Q}$ we denote their *total variation distance* by $d_{\mathrm{TV}}(\mathbb{P}, \mathbb{Q}) = \max_{\text{event } E} |\mathbb{P}(E) - \mathbb{Q}(E)|$.

**Notation.** Our bound depends on many parameters of the problem, specified below. Additional parameters (bounds) are specified in the assumptions stated below.

| | | | |
|---|---|---|---|
| $n$ | horizon, i.e. number of rounds | $m$ | $:= \lceil \log_2(n) + 1 \rceil$ |
| $s, t$ | indices of rounds | $X_{<t}$ | $\in \mathbb{R}^{(t-1) \times d}$, with $X_{<t,s} = \boldsymbol{x}_s^\mathsf{T}$ for $s < t$ |
| $d$ | dimensionality of action space | $G_t$ | Gram matrix of the actions: $X_{<t}^\mathsf{T} X_{<t}$ |
| $\mathcal{D}_t$ | $\subset \mathbb{R}^d$; decision set at round $t$ | $H_t$ | regularizer at round $t$ |
| $\boldsymbol{x}_t$ | $\in \mathcal{D}_t$; action at round $t$ | $\boldsymbol{y}_{<t}$ | vector of rewards up to round $t-1$ |
| $y_t$ | $\in \mathbb{R}$; reward at round $t$ | $\boldsymbol{u}_t$ | action-reward product: $X_{<t}^\mathsf{T} \boldsymbol{y}_{<t}$ |
| $\boldsymbol{\theta}^*$ | $\in \mathbb{R}^d$; unknown parameter vector | $\boldsymbol{h}_t$ | perturbation of $\boldsymbol{u}_t$ |

## 3 Linear UCB with Changing Regularizers

In this section we introduce and analyze a variation of the well-studied LinUCB algorithm, an application of the Upper Confidence Bound (UCB) idea to stochastic linear bandits [13, 29, 1]. At every round $t$, LinUCB constructs a *confidence set* $\mathcal{E}_t$ that contains the unknown parameter vector $\boldsymbol{\theta}^*$ with high probability. It then computes an upper confidence bound on the reward of each action in the decision set $\mathcal{D}_t$, and "optimistically" chooses the action with the highest UCB: $\boldsymbol{x}_t \leftarrow \arg\max_{\boldsymbol{x} \in \mathcal{D}_t} \mathrm{UCB}_t(\boldsymbol{x})$, where $\mathrm{UCB}_t(\boldsymbol{x}) := \max_{\boldsymbol{\theta} \in \mathcal{E}_t} \langle \boldsymbol{\theta}, \boldsymbol{x} \rangle$. We assume the rewards are linear with added subgaussian noise (i.e., $y_s = \langle \boldsymbol{\theta}^*, \boldsymbol{x}_s \rangle + \eta_s$ for $s < t$), so it is natural to center the confidence set $\mathcal{E}_t$ on the (regularized) linear regression estimate:

$$\hat{\boldsymbol{\theta}}_t := \arg\min_{\hat{\boldsymbol{\theta}} \in \mathbb{R}^d} \|X_{<t}\hat{\boldsymbol{\theta}} - \boldsymbol{y}_{<t}\|^2 + \|\hat{\boldsymbol{\theta}}\|_{H_t}^2 = (G_t + H_t)^{-1} X_{<t}^\mathsf{T} \boldsymbol{y}_{<t}. \qquad \text{where } G_t := X_{<t}^\mathsf{T} X_{<t}$$

The matrix $V_t := G_t + H_t \in \mathbb{R}^{d \times d}$ is a regularized version of the Gram matrix $G_t$. Whenever the learner chooses an action vector $\boldsymbol{x} \in \mathcal{D}_t$, the corresponding reward gives it some information about the projection of $\boldsymbol{\theta}^*$ onto $\boldsymbol{x}$. In other words, the estimate $\hat{\boldsymbol{\theta}}$ is probably closer to $\boldsymbol{\theta}^*$ along the directions where many actions have been taken. This motivates the use of ellipsoidal confidence sets that are smaller in such directions, inversely corresponding to the eigenvalues of $G_t$ (or $V_t$). The ellipsoid is uniformly scaled by $\beta_t$ to achieve the desired confidence level, as prescribed by Proposition 4.

$$\mathcal{E}_t := \{\boldsymbol{\theta} \in \mathbb{R}^d \mid \|\boldsymbol{\theta} - \hat{\boldsymbol{\theta}}_t\|_{V_t} \leq \beta_t\}, \qquad \text{for which } \mathrm{UCB}_t(\boldsymbol{x}) = \langle \hat{\boldsymbol{\theta}}, \boldsymbol{x} \rangle + \beta_t \|\boldsymbol{x}\|_{V_t^{-1}}. \qquad (1)$$

Just as the changing regularizer $H_t$ perturbs the Gram matrix $G_t$, our algorithm allows for the vector $\boldsymbol{u}_t := X_{<t}^\mathsf{T} \boldsymbol{y}_{<t}$ to be perturbed by $\boldsymbol{h}_t$ to get $\tilde{\boldsymbol{u}}_t := \boldsymbol{u}_t + \boldsymbol{h}_t$. The estimate $\hat{\boldsymbol{\theta}}_t$ is replaced by $\tilde{\boldsymbol{\theta}}_t := V_t^{-1} \tilde{\boldsymbol{u}}_t$.

---
**Algorithm 1** Linear UCB with Changing Perturbations
---
**Initialize:** $G_1 \leftarrow \boldsymbol{0}_{d \times d}$, $u_1 \leftarrow \boldsymbol{0}_d$.
**for** each round $t = 1, 2, \ldots, n$ **do**
    Receive $\mathcal{D}_t \leftarrow$ decision set $\subset \mathbb{R}^d$.
    Receive regularized $V_t \leftarrow G_t + H_t$ and perturbed $\tilde{\boldsymbol{u}}_t \leftarrow \boldsymbol{u}_t + \boldsymbol{h}_t$
    Compute regressor $\tilde{\boldsymbol{\theta}}_t \leftarrow V_t^{-1} \tilde{\boldsymbol{u}}_t$
    Compute confidence-set bound $\beta_t$ based on Proposition 4.
    Pick action $\boldsymbol{x}_t \leftarrow \arg\max_{\boldsymbol{x} \in \mathcal{D}_t} \langle \tilde{\boldsymbol{\theta}}_t, \boldsymbol{x} \rangle + \beta_t \|\boldsymbol{x}\|_{V_t^{-1}}$.
    Observe $y_t \leftarrow$ reward for action $\boldsymbol{x}_t$
    Update: $G_{t+1} \leftarrow G_t + \boldsymbol{x}_t \boldsymbol{x}_t^\mathsf{T}, \quad \boldsymbol{u}_{t+1} \leftarrow \boldsymbol{u}_t + \boldsymbol{x}_t y_t$
**end for**
---

Our analysis relies on the following assumptions about the environment and algorithm:

**Assumptions** (Algorithm 1)**.** For all rounds $t = 1, \ldots, n$ and actions $\boldsymbol{x} \in \mathcal{D}_t$:

1. Bounded action set: $\|\boldsymbol{x}\| \le L$.
2. Bounded *mean* reward: $|\langle \boldsymbol{\theta}^*, \boldsymbol{x}\rangle| \le B$ with $B \ge 1$.[2]
3. Bounded target parameter: $\|\boldsymbol{\theta}^*\| \le S$.
4. All regularizers are PSD $\boldsymbol{H}_t \succeq 0$.
5. $y_t = \langle \boldsymbol{\theta}^*, \boldsymbol{x}_t\rangle + \eta_t$ where $\eta_t$ is $\sigma^2$-conditionally subgaussian on previous actions and rewards, i.e.: $\mathbb{E}[\exp(\lambda\eta_t) \mid \boldsymbol{x}_1, y_1, \dots, \boldsymbol{x}_{t-1}, y_{t-1}, \boldsymbol{x}_t] \le \exp(\lambda^2\sigma^2/2)$, for all $\lambda \in \mathbb{R}$.
6. Strongest bound on both the action and the reward: $\|\boldsymbol{x}_t\|^2 + y_t^2 \le \tilde{L}^2$ for all rounds $t$. In particular, if $\|\boldsymbol{x}_t\| \le L$ (Assumption 1) and the rewards are bounded: $|y_t| \le \tilde{B}$ (not just their means as in Assumption 2), we can set $\tilde{L}^2 = L^2 + \tilde{B}^2$.

Assumptions 1 and 2 aren't required to have good pseudo-regret bounds, they merely simplify the bounds on the confidence set (see Proposition 4 and Section 3.1). Assumption 6 is not required at all for now, it is only used for joint differential privacy in Section 4.

To fully describe Algorithm 1 we need to specify how to compute the confidence-set bounds $(\beta_t)_t$. On the one hand, these bounds have to be accurate — the confidence set $\mathcal{E}_t$ should contain the unknown $\boldsymbol{\theta}^*$; on the other hand, the larger they are, the larger the regret bounds we obtain. In other words, the $\beta_t$ should be as small as possible subject to being accurate.

**Definition 2** (Accurate $(\beta_t)_t$). A sequence $(\beta_t)_{t=1}^n$ is $(\alpha, n)$-*accurate* for $(\boldsymbol{H}_t)_{t=1}^n$ and $(\boldsymbol{h}_t)_{t=1}^n$ if, with probability at least $1 - \alpha$, it satisfies $\|\boldsymbol{\theta}^* - \bar{\boldsymbol{\theta}}_t\|_{V_t} \le \beta_t$ for all rounds $t = 1, \dots, n$ simultaneously.

We now argue that three parameters are the key to establishing accurate confidence-set bounds $(\beta_t)_{t=1}^n$ — taking into account the noise in the setting *and* the noise added by a changing $\boldsymbol{H}_t$ and $\boldsymbol{h}_t$.

**Definition 3** (Accurate $\rho_{\min}$, $\rho_{\max}$, and $\gamma$). The bounds $0 < \rho_{\min} \le \rho_{\max}$ and $\gamma$ are $(\alpha/2n)$-*accurate* for $(\boldsymbol{H}_t)_{t=1}^n$ and $(\boldsymbol{h}_t)_{t=1}^n$ if for each round $t$:

$$\|\boldsymbol{H}_t\| \le \rho_{\max}, \qquad \|\boldsymbol{H}_t^{-1}\| \le 1/\rho_{\min}, \qquad \|\boldsymbol{h}_t\|_{\boldsymbol{H}_t^{-1}} \le \gamma; \qquad \text{with probability at least } 1 - \alpha/2n.$$

**Proposition 4** (Calculating $\beta_t$). *Suppose Assumptions 3 to 5 hold and let $\rho_{\min}$, $\rho_{\max}$, and $\gamma$ be $(\alpha/2n)$-accurate for some $\alpha \in (0, 1)$ and horizon $n$. Then $(\beta_t)_{t=1}^n$ is $(\alpha, n)$-accurate where*

$$\beta_t := \sigma\sqrt{2\log(2/\alpha) + \log(\det V_t) - d\log(\rho_{\min})} + S\sqrt{\rho_{\max}} + \gamma$$

$$\le \sigma\sqrt{2\log(2/\alpha) + d\log\left(\frac{\rho_{\max}}{\rho_{\min}} + \frac{tL^2}{d\rho_{\min}}\right)} + S\sqrt{\rho_{\max}} + \gamma. \qquad \text{(if Assumption 1 also holds)}$$

### 3.1 Regret Bounds

We now present bounds on the maximum regret of Algorithm 1. However, due to space constraints, we defer an extensive discussion of the proof techniques used and the significance of the results to Section B in the supplementary material. The proofs are based on those of Abbasi-Yadkori et al. [1], who analyzed LinUCB with constant regularizers. On one hand, our changes are mostly technical; however, it turns out that various parts of the proof diverge and now depend on $\rho_{\max}$ and $\rho_{\min}$; tracing them all is somewhat involved. It is an interesting question to establish similar bounds using known results only as a black box; we were not able to accomplish this.

**Theorem 5** (Regret of Algorithm 1). *Suppose Assumptions 1 to 5 hold and the $\beta_t$ are as given by Proposition 4. Then with probability at least $1 - \alpha$ the pseudo-regret of Algorithm 1 is bounded by*

$$\widehat{R}_n \le B\sqrt{8n}\left[\sigma\left(2\log(\tfrac{2}{\alpha}) + d\log\left(\tfrac{\rho_{\max}}{\rho_{\min}} + \tfrac{nL^2}{d\rho_{\min}}\right)\right) + (S\sqrt{\rho_{\max}} + \gamma)\sqrt{d\log\left(1 + \tfrac{nL^2}{d\rho_{\min}}\right)}\right] \qquad (2)$$

**Theorem 6** (Gap-Dependent Regret of Algorithm 1). *Suppose Assumptions 1 to 5 hold and the $\beta_t$ are as given by Proposition 4. If the optimal actions in every decision set $\mathcal{D}_t$ are separated from the sub-optimal actions by a reward gap of at least $\Delta$, then with probability at least $1 - \alpha$ the pseudo-regret of Algorithm 1 satisfies*

$$\widehat{R}_n \le \frac{8B}{\Delta}\left[\sigma\left(2\log(\tfrac{2}{\alpha}) + d\log\left(\tfrac{\rho_{\max}}{\rho_{\min}} + \tfrac{nL^2}{d\rho_{\min}}\right)\right) + (S\sqrt{\rho_{\max}} + \gamma)\sqrt{d\log\left(1 + \tfrac{nL^2}{d\rho_{\min}}\right)}\right]^2 \qquad (3)$$

# 4 Linear UCB with Joint Differential Privacy

Notice that Algorithm 1 uses its history of actions and rewards up to round $t$ only via the confidence set $\mathcal{E}_t$, which is to say via $V_t$ and $\tilde{u}_t$, which are perturbations of the Gram matrix $G_t$ and the vector $u_t := X_{<t}^\mathsf{T} y_{<t}$, respectively; these also determine $\beta_t$. By recording this history with differential privacy, we obtain a Linear UCB algorithm that is jointly differentially private (Definition 1) because it simply post-processes $G_t$ and $u_t$.

**Claim 7** (see Dwork and Roth [15, Proposition 2.1]). *If the sequence $(V_t, \tilde{u}_t)_{t=1}^{n-1}$ is $(\varepsilon, \delta)$-differentially private with respect to $(x_t, y_t)_{t=1}^{n-1}$, then Algorithm 1 is $(\varepsilon, \delta)$-jointly differentially private.*

*Remark* 1. Algorithm 1 is only jointly differentially private even though the history maintains full differential privacy — its action choice depends not only on the past contexts $c_s$ ($s < t$, via the differentially private $X_{<t}$) but also on the current context $c_t$ via the decision set $\mathcal{D}_t$. This use of $c_t$ is *not* differentially private, as it is revealed by the algorithm's chosen $x_t$.

Rather than applying the tree-based algorithm separately to $G_t$ and $u_t$, we aggregate both into the single matrix $M_t \in \mathbb{R}^{(d+1)\times(d+1)}$, which we now construct. Define $A := \begin{bmatrix} X_{1:n} & y_{1:n} \end{bmatrix} \in \mathbb{R}^{n\times(d+1)}$, with $A_t$ holding the top $t-1$ rows of $A$ (and $A_1 = \mathbf{0}_{1\times(d+1)}$). Now let $M_t := A_t^\mathsf{T} A_t$ — then the top-left $d \times d$ sub-matrix of $M_t$ is the Gram matrix $G_t$ and the first $d$ entries of its last column are $u_t$. Furthermore, since $M_{t+1} = M_t + \begin{bmatrix} x_t^\mathsf{T} & y_t \end{bmatrix}^\mathsf{T} \begin{bmatrix} x_t^\mathsf{T} & y_t \end{bmatrix}$, the tree-based algorithm for private cumulative sums can be used to maintain a private estimation of $M_t$ using additive noise, releasing $M_t + N_t$. The top-left $d \times d$ sub-matrix of $N_t$ becomes $H_t$ and the first $d$ entries of its last column become $h_t$. Lastly, to have a private estimation of $M_t$, Assumption 6 must hold.

Below we present two techniques for maintaining (and updating) the private estimations of $M_t$. As mentioned in Section 1.3, the key component of our technique is the tree-based algorithm, allowing us to estimate $M_t$ using at most $m := 1 + \lceil \log_2 n \rceil$ noisy counters. In order for the entire tree-based algorithm to be $(\varepsilon, \delta)$-differentially private, we add noise to each node in the tree so that each noisy count on its own preserves $(\varepsilon/\sqrt{8m\ln(2/\delta)}, \delta/2m)$-differential privacy. Thus in each day, the noise $N_t$ that we add to $M_t$ comes from the sum of at most $m$ such counters.

## 4.1 Differential Privacy via Wishart Noise

First, we instantiate the tree-based algorithm with noise from a suitably chosen Wishart distribution $\mathcal{W}_{d+1}(V, k)$, which is the result of sampling $k$ independent $(d+1)$-dimensional Gaussians from $\mathcal{N}(\mathbf{0}_{d+1}, V)$ and computing their Gram matrix.

**Theorem 8** (Theorem 4.1 [30]). *Fix positive $\varepsilon_0$ and $\delta_0$. If the $L_2$-norm of each row in the input is bounded by $\tilde{L}$ then releasing the input's Gram matrix with added noise sampled from $\mathcal{W}_{d+1}(\tilde{L}^2 I, k_0)$ is $(\varepsilon_0, \delta_0)$-differentially private, provided $k_0 \geq d + 1 + 28\varepsilon_0^{-2}\ln(4/\delta_0)$.*

Applying this guarantee to our setting, where each count needs to preserve $(\varepsilon/\sqrt{8m\ln(2/\delta)}, \delta/2m)$-differential privacy, it suffices to sample a matrix from $\mathcal{W}_{d+1}(\tilde{L}I, k)$ with $k := d + 1 + \lceil 224m\varepsilon^{-2}\ln(8m/\delta)\ln(2/\delta) \rceil$. Moreover, the sum of $m$ independent samples from the Wishart distribution is a noise matrix $N_t \sim \mathcal{W}_{d+1}(\tilde{L}^2 I, mk)$.[3] Furthermore, consider the regularizers $H_t$ and $h_t$ derived from $N_t$ (the top-left submatrix and the right-most subcolumn resp.) — $H_t$ has distribution $\mathcal{W}_d(\tilde{L}^2 I, mk)$, and each entry of $h_t$ is the dot-product of two multivariate Gaussians. Knowing their distribution, we can infer the accurate bounds required for our regret bounds. Furthermore, since the Wishart noise has eigenvalues that are fairly large, we consider a post-processing of the noise matrix — shifting it by $-cI$ with

$$c := \tilde{L}^2\left(\sqrt{mk} - \sqrt{d} - \sqrt{2\ln(8n/\alpha)}\right)^2 - 4\tilde{L}^2\sqrt{mk}\left(\sqrt{d} + \sqrt{2\ln(8n/\alpha)}\right) \tag{4}$$

making the bounds we require smaller than without the shift. The derivations are deferred to Section C of the supplementary material.

**Proposition 9.** *Fix any $\alpha > 0$. If for each $t$ the $\boldsymbol{H}_t$ and $\boldsymbol{h}_t$ are generated by the tree-based algorithm with Wishart noise $\mathcal{W}_{d+1}(\tilde{L}^2\boldsymbol{I}, k)$, then the following are $(\alpha/2n)$-accurate bounds:*

$$\rho_{\min} = \tilde{L}^2\big(\sqrt{mk} - \sqrt{d} - \sqrt{2\ln(8n/\alpha)}\big)^2,$$

$$\rho_{\max} = \tilde{L}^2\big(\sqrt{mk} + \sqrt{d} + \sqrt{2\ln(8n/\alpha)}\big)^2,$$

$$\gamma = \tilde{L}\big(\sqrt{d} + \sqrt{2\ln(2n/\alpha)}\big).$$

*Moreover, if we use the shifted regularizer $\boldsymbol{H}_t' := \boldsymbol{H}_t - c\boldsymbol{I}$ with $c$ as given in Eq. (4), then the following are $(\alpha/2n)$-accurate bounds:*

$$\rho_{\min}' = 4\tilde{L}^2\sqrt{mk}\big(\sqrt{d} + \sqrt{2\ln(8n/\alpha)}\big),$$

$$\rho_{\max}' = 8\tilde{L}^2\sqrt{mk}\big(\sqrt{d} + \sqrt{2\ln(8n/\alpha)}\big),$$

$$\gamma' = \tilde{L}\sqrt{\sqrt{mk}\big(\sqrt{d} + \sqrt{2\ln(2n/\alpha)}\big)}.$$

Plugging these into Theorems 5 and 6 gives us the following upper bounds on pseudo-regret.

**Corollary 10.** *Algorithm 1 with $\boldsymbol{H}_t$ and $\boldsymbol{h}_t$ generated by the tree-based mechanism with each node adding noise independently from $\mathcal{W}_{d+1}((L^2 + \tilde{B}^2)\boldsymbol{I}, k)$ and then subtracting $c\boldsymbol{I}$ using Eq. (4), we get a pseudo-regret bound of*

$$O\left(B\sqrt{n}\left[\sigma\big(\log(1/\alpha) + d\log(n)\big) + S\tilde{L}\sqrt{d}\log(n)^{3/4}(d^{1/4} + \varepsilon^{-1/2}\log(1/\delta)^{1/4})(d^{1/4} + \log(n/\alpha)^{1/4})\right]\right)$$

*in general, and a gap-dependent pseudo-regret bound of*

$$O\left(\frac{B}{\Delta}\left[\sigma\big(\log(1/\alpha) + d\log(n)\big) + S\tilde{L}\sqrt{d}\log(n)^{3/4}(d^{1/4} + \varepsilon^{-1/2}\log(1/\delta)^{1/4})(d^{1/4} + \log(n/\alpha)^{1/4})\right]^2\right)$$

## 4.2 Differential Privacy via Additive Gaussian Noise

Our second alternative is to instantiate the tree-based algorithm with symmetric Gaussian noise: sample $\boldsymbol{Z}' \in \mathbb{R}^{(d+1)\times(d+1)}$ with each $\boldsymbol{Z}_{i,j}' \sim \mathcal{N}(0, \sigma_{\text{noise}}^2)$ i.i.d. and symmetrize to get $\boldsymbol{Z} = (\boldsymbol{Z}' + \boldsymbol{Z}'^{\mathsf{T}})/\sqrt{2}$.[4] Recall that each datum has a bounded $L_2$-norm of $\tilde{L}$, hence a change to a single datum may alter the Frobenius norm of $\boldsymbol{M}_t$ by $\tilde{L}^2$. It follows that in order to make sure each node in the tree-based algorithm preserves $(\varepsilon/\sqrt{8m\ln(2/\delta)}, \delta/2)$-differential privacy,[5] the variance in each coordinate must be $\sigma_{\text{noise}}^2 = 16m\tilde{L}^4\ln(4/\delta)^2/\varepsilon^2$. When all entries on $\boldsymbol{Z}$ are sampled from $\mathcal{N}(0, 1)$, known concentration results [32] on the top singular value of $\boldsymbol{Z}$ give that $\mathbb{P}[\|\boldsymbol{Z}\| > (4\sqrt{d+1} + 2\ln(2n/\alpha))] < \alpha/2n$. Note however that in each day $t$ the noise $\boldsymbol{N}_t$ is the sum of $\leq m$ such matrices, thus the variance of each coordinate is $m\sigma_{\text{noise}}^2$. The top-left $(d \times d)$-submatrix of $\boldsymbol{N}_t$ has operator norm of at most

$$\Upsilon := \sigma_{\text{noise}}\sqrt{2m}\big(4\sqrt{d} + 2\ln(2n/\alpha)\big) = \sqrt{32}m\tilde{L}^2\ln(4/\delta)\big(4\sqrt{d} + 2\ln(2n/\alpha)\big)/\varepsilon.$$

However, it is important to note that the result of adding Gaussian noise may not preserve the PSD property of the noisy Gram matrix. To that end, we ought to shift $\boldsymbol{N}_t$ by some $c\boldsymbol{I}$ in order to make sure that we maintain strictly positive eigenvalues throughout the execution of the algorithm. Since the bounds in Theorems 5 and 6 mainly depend on $\sqrt{\rho_{\max}} + \gamma$, we choose the shift-magnitude to be $2\Upsilon\boldsymbol{I}$. This makes $\rho_{\max} = 3\Upsilon$ and $\rho_{\min} = \Upsilon$ and as a result $\|\boldsymbol{h}_t\|_{\boldsymbol{H}_t^{-1}} \leq \sqrt{\Upsilon^{-1}}\|\boldsymbol{h}_t\|$, which we can bound using standard concentration bounds on the $\chi^2$-distribution (see Claim 17). This culminates in the following bounds.

**Proposition 11.** *Fix any $\alpha > 0$. Given that for each $t$ the regularizers $\boldsymbol{H}_t, \boldsymbol{h}_t$ are taken by applying the tree-based algorithm with symmetrized shifted Gaussian noise whose entries are sampled i.i.d. from $\mathcal{N}(0, \sigma_{\text{noise}}^2)$, then the following $\rho_{\min}$, $\rho_{\max}$, and $\gamma$ are $(\alpha/2n)$-accurate bounds:*

$$\rho_{\min} = \Upsilon, \quad \rho_{\max} = 3\Upsilon, \quad \gamma = \sigma_{\text{noise}}\sqrt{\Upsilon^{-1}m}\big(\sqrt{d} + \sqrt{2\ln(2n/\alpha)}\big) \leq \sqrt{m\tilde{L}^2\big(\sqrt{d} + 2\ln(2n/\alpha)\big)/(\sqrt{2}\varepsilon)}$$

Note how this choice of shift indeed makes both $\rho_{\max}$ and $\gamma^2$ roughly on the order of $O(\Upsilon)$.

The end result is that for each day $t$, $\boldsymbol{h}_t$ is given by summing at most $m$ $d$-dimensional vectors whose entries are sampled i.i.d. from $\mathcal{N}(0, \sigma_{\text{noise}}^2)$; the symmetrization doesn't change the distribution of each coordinate. The matrix $\boldsymbol{H}_t$ is given by: (i) summing at most $m$ matrices whose entries are sampled i.i.d. from $\mathcal{N}(0, \sigma_{\text{noise}}^2)$; (ii) symmetrizing the result as shown above; and (iii) adding $2\Upsilon\boldsymbol{I}$. This leads to a bound on the regret of Algorithm 1 with the tree-based algorithm using Gaussian noise.

**Corollary 12.** *Applying Algorithm 1 where the regularizers $\boldsymbol{H}_t$ and $\boldsymbol{h}_t$ are derived by applying the tree-based algorithm where each node holds a symmetrized matrix whose entries are sampled i.i.d. from $\mathcal{N}(0, \sigma_{\text{noise}}^2)$ and adding $2\Upsilon\boldsymbol{I}$, we get a regret bound of*

$$O\left(B\sqrt{n}\left(\sigma(d\log(n) + \log(1/\alpha)) + S\tilde{L}\log(n)\sqrt{d(\sqrt{d} + \ln(n/\alpha))\ln(1/\delta)/\varepsilon}\right)\right)$$

*in general, and a gap-dependent pseudo-regret bound of*

$$O\left(\frac{B}{\Delta}\left(\sigma(d\log(n) + \log(1/\alpha)) + S\tilde{L}\log(n)\sqrt{d(\sqrt{d} + \ln(n/\alpha))\ln(1/\delta)/\varepsilon}\right)^2\right)$$

## 5 Lower Bounds

In this section, we present lower bounds for two versions of the problem we deal with in this work. The first, and probably the more obvious of the two, deals with the impossibility of obtaining sub-linear regret for the contextual bandit problem with the standard notion of differential privacy (under continual observation). Here, we assume user $t$ provides a context $c_t$ which actually determines the mapping of the arms into feature vectors: $\phi(c_t, a) \in \mathbb{R}^d$. The sequence of choice thus made by the learner is $a_1, \ldots, a_n \in \mathcal{A}^n$ which we aim to keep private. The next claim, whose proof is deferred to Section C in the supplementary material, shows that this is impossible without effectively losing any reasonable notion of utility.

**Claim 13.** *For any $\varepsilon < \ln(2)$ and $\delta < 0.25$, any $(\varepsilon, \delta)$-differentially private algorithm $A$ for the contextual bandit problem must incur pseudo-regret of $\Omega(n)$.*

The second lower bound we show is more challenging. We show that any $\varepsilon$-differentially private algorithm for the classic MAB problem must incur *an additional* pseudo-regret of $\Omega(k\log(n)/\epsilon)$ on top of the standard (non-private) regret bounds. We consider an instance of the MAB where the leading arm is $a^1$, the rewards are drawn from a distribution over $\{-1, 1\}$, and the gap between the means of arm $a^1$ and arm $a \neq a^1$ is $\Delta_a$. Simple calculation shows that for such distributions, the total-variation distance between two distributions whose means are $\mu$ and $\mu - \Delta$ is $\Delta/2$. Fix $\Delta_2, \Delta_3, \ldots, \Delta_k$ as some small constants, and we now argue the following.

**Claim 14.** *Let $A$ be any $\varepsilon$-differentially private algorithm for the MAB problems with $k$ arms whose expected regret is at most $n^{3/4}$. Fix any arm $a \neq a^1$, whose difference between it and the optimal arm $a^1$ is $\Delta_a$. Then, for sufficiently large $ns$, $A$ pulls arm $a$ at least $\log(n)/100\varepsilon\Delta_a$ many times w.p. $\geq 1/2$.*

We comment that the bound $n^{3/4}$ was chosen arbitrarily, and we only require a regret upper bound of $n^{1-c}$ for some $c > 0$. Of course, we could have used standard assumptions, where the regret is asymptotically smaller than *any* polynomial; or discuss algorithms of regret $\tilde{O}(\sqrt{n})$ (best minimax regret). Aiming to separate the standard lower-bounds on regret from the private bounds, we decided to use $n^{3/4}$. As an immediate corollary we obtain the following *private* regret bound:

**Corollary 15.** *The expected pseudo-regret of any $\varepsilon$-differentially private algorithm for the MAB is $\Omega(k\log(n)/\varepsilon)$. Combined with the non-private bound of $\Omega\left(\sum_{a\neq a^1} \log(n)/\Delta_a\right)$ we get that the private regret bound is the* max *of the two terms, i.e.: $\Omega\left(k\log(n)/\varepsilon + \sum_{a\neq a^1} \log(n)/\Delta_a\right)$.*

*Proof.* Based on Claim 14, the expected pseudo-regret is at least $\sum\limits_{a\neq a^1} \frac{\Delta_a \log(n)}{200\varepsilon\Delta_a} = \frac{(k-1)\log(n)}{200\varepsilon}$. □

*Proof of Claim 14.* Fix arm $a$. Let $\bar{P}$ be the vector of the $k$-probability distributions associated with the $k$ arms. Denote $E$ as the event that arm $a$ is pulled $< \log(n)/100\varepsilon\Delta_a := t_a$ many times. Our goal is to show that $\mathbb{P}_{A; \text{ rewards}\sim\bar{P}}[E] < 1/2$.

To that end, we postulate a different distribution for the rewards of arm $a$ — a new distribution whose mean is *greater* by $\Delta_a$ than the mean reward of arm $a^1$. The total-variation distance between the given distribution and the postulated distribution is $\Delta_a$. Denote $\bar{Q}$ as the vector of distributions of

arm-rewards (where only $P_a \neq Q_a$). We now argue that should the rewards be drawn from $\bar{Q}$, then the event $E$ is very unlikely: $\mathbb{P}_{A; \text{ rewards}\sim\bar{Q}}[E] \leq 2n^{-1/4}/\Delta_a$. Indeed, the argument is based on a standard Markov-like argument: the expected pseudo-regret of $A$ is at most $n^{3/4}$, yet it is at least $\mathbb{P}_{A; \text{ rewards}\sim\bar{Q}}[E] \cdot (n - t_a)\Delta_a \geq (n\Delta_a/2)\mathbb{P}_{A; \text{ rewards}\sim\bar{Q}}[E]$, for sufficiently large $n$.

We now apply a beautiful result of Karwa and Vadhan [22, Lemma 6.1], stating that the "effective" group privacy between the case where the $n$ datums of the inputs are drawn i.i.d. from either distribution $P$ or from distribution $Q$ is proportional to $\varepsilon n \cdot d_{\text{TV}}(P, Q)$. In our case, the key point is that we only consider this change *under the event $E$*, thus the number of samples we need to redraw from the distribution $P_a$ rather than $Q_a$ is strictly smaller than $t_a$, and the elegant coupling argument of [22] reduces it to $6\Delta_a \cdot t_a$. To better illustrate the argument, consider the coupling argument of [22] as an oracle $O$. The oracle generates a collection of precisely $t_a$ *pairs* of points, the left ones are i.i.d. samples from $P_a$ and the right ones are i.i.d. samples from $Q_a$, and, in expectation, in $(1 - \Delta_a)$ fraction of the pairs the right- and the left-samples are identical. Whenever the learner $A$ pulls arm $a$ it makes an oracle call to $O$, and depending on the environment (whether the distribution of rewards is $\bar{P}$ or $\bar{Q}$) $O$ provides either a fresh left-sample or a right-sample. Moreover, suppose there exists a counter $C$ that stands between $A$ and $O$, and in case $O$ runs out of examples then $C$ routes $A$'s oracle calls to a different oracle. Now, Karwa and Vadhan [22, Lemma 6.1] assures that the probability of the event "$C$ never re-routes the requests" happens with similar probability under $P$ or under $Q$, different only up to a multiplicative factor of $\exp(\epsilon\Delta_a t_a)$. And seeing as the event "$C$ never re-routes the requests" is quite unlikely when $O$ only provides right-samples (from $\bar{Q}$), it is also fairly unlikely when $O$ only provides left-samples (from $\bar{P}$).

Formally, we conclude the proof by applying the result of [22] to infer that $\mathbb{P}_{A; \text{ rewards}\sim\bar{P}}[E] \leq \exp(6\varepsilon\Delta_a t_a)\mathbb{P}_{A; \text{ rewards}\sim\bar{Q}}[E] \leq \exp(0.06\log(n)) \cdot \frac{2}{\Delta_a}n^{-1/4} = n^{-0.19}\frac{2}{\Delta_a} \leq 1/2$ for sufficiently large $n$s, proving the required claim. $\square$

## Acknowledgements

We gratefully acknowledge the Natural Sciences and Engineering Research Council of Canada (NSERC) for supporting R.S. with the Alexander Graham Bell Canada Graduate Scholarship and O.S. with grant #2017–06701. R.S. was also supported by Alberta Innovates and O.S. is also an unpaid collaborator on NSF grant #1565387.

## Footnotes

[1]The pseudo-regret ignores the stochasticity of the reward but not the resulting randomness in the learner's choice of actions. It equals the regret in expectation, but is more amenable to high-probability bounds such as ours. In particular, in some cases we can achieve polylog($n$) bounds on pseudo-regret because, unlike regret, it doesn't have added regret noise of variance $\Omega(n)$.

[2] See Remark 2 preceding the proof of Lemma 21 in Section B of the supplementary material for a discussion as to bounding $B$ by 1 from below.

[3]Intuitively, we merely concatenate the $m$ batches of multivariate Gaussians sampled in the generation of each of the $m$ Wishart noises.

[4]This increases the variance along the diagonal entries beyond the noise magnitude required to preserve privacy, but only by a constant factor of 2.

[5]We use here the slightly better bounds for the composition of Gaussian noise based on zero-Concentrated DP [9].

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
