[Supplementary Material]

# Supplementary Material

## A   Additional Background Information

### A.1   Differential Privacy

In the offline setting, a dataset $D$ is a $n$-tuple of elements from some universe $\mathcal{U}$. Two datasets are called neighbors if they differ just on a single element. An algorithm $A$ is said to be $(\varepsilon, \delta)$-differentially private if for any pair of neighboring datasets $D$ and $D'$ and any subset of possible outputs $\mathcal{S}$ we have that $\mathbb{P}\{A(D) \in \mathcal{S}\} \le e^{\varepsilon} \mathbb{P}\{A(D') \in \mathcal{S}\} + \delta$. A common technique [16] for approximating the value of a query $f$ on dataset $D$ is to first find its $L_2$-sensitivity, $GS_2 := \max_{D, D' \text{ neighboring}} \|f(D) - f(D')\|_2$, and then add zero-mean Gaussian noise of variance $2GS_2^2 \ln(2/\delta)/\varepsilon^2$.

### A.2   The Tree-Based Mechanism

Assume for simplicity that $n = 2^i$ for some positive integer $i$. Let $T$ be a complete binary tree with its leaf nodes being $l_1, \dots, l_n$. Each internal node $x \in T$ stores the sum of all the leaf nodes in the tree rooted at $x$. First notice that one can compute any partial sum $\sum_{j=1}^{i} l_i$ using at most $m := \lceil \log_2(n) + 1 \rceil$ nodes of $T$. Second, notice that for any two neighboring data sequences $D$ and $D'$ the partial sums stored in $T$ differ on at most $m$ nodes. Thus, if the count in each node preserves $(\varepsilon_0, \delta_0)$-differential privacy, using the advanced composition of Dwork et al. [14] we get that the entire algorithm is $\left( O(m\varepsilon_0^2 + \varepsilon_0 \sqrt{2m \ln(1/\delta')}), m\delta_0 + \delta' \right)$-differentially private. Alternatively, to make sure the entire tree is $(\varepsilon, \delta)$-differentially private, it suffices to set $\varepsilon_0 = \varepsilon/\sqrt{8m \ln(2/\delta)}$ and $\delta_0 = \delta/2m$ (with $\delta' = \delta/2$).

### A.3   Useful Facts.

In this work, we repeatedly apply the following facts about PSD matrices, the Gaussian distribution, the $\chi^2$ distribution and the Wishart distribution.

**Claim 16** (36, Theorem 7.8). *If $A \succeq B \succeq 0$, then*

    *1.* $\operatorname{rank}(A) \ge \operatorname{rank}(B)$
    *2.* $\det A \ge \det B$
    *3.* $B^{-1} \succeq A^{-1}$ *if $A$ and $B$ are nonsingular.*

**Claim 17** (Corollary to Lemma 1, 25, p. 1325). *If $U \sim \chi^2(d)$ and $\alpha \in (0,1)$,*

$$\mathbb{P}\left( U \ge d + 2\sqrt{d \ln \tfrac{1}{\alpha}} + 2 \ln \tfrac{1}{\alpha} \right) \le \alpha, \qquad \mathbb{P}\left( U \le d - 2\sqrt{d \ln \tfrac{1}{\alpha}} \right) \le \alpha.$$

**Claim 18** (Adaptation of 35, Corollary 5.35). *Let $A$ be an $n \times d$ matrix whose entries are independent standard normal variables. Then for every $\alpha \in (0,1)$, with probability at least $1 - \alpha$ it holds that*

$$\sigma_{\min}(A), \sigma_{\max}(A) \in \sqrt{n} \pm (\sqrt{d} + \sqrt{2 \ln(2/\alpha)})$$

*with $\sigma_{\min}(A)$ and $\sigma_{\max}(A)$ denoting the smallest- and largest singular values of $A$ resp.*

**Claim 19** (30, Lemma A.3). *Fix $\alpha \in (0, 1/e)$ and let $W \sim \mathcal{W}_d(V, k)$ with $\sqrt{m} > \sqrt{d} + \sqrt{2 \ln(2/\alpha)}$. Then, denoting the $j$-t largest eigenvalue of $W$ as $\sigma_j(W)$, with probability at least $1 - \alpha$ it holds that for every $j = 1, 2, \dots, d$:*

$$\sigma_j(W) \in \left( \sqrt{m} \pm \left( \sqrt{d} + \sqrt{2 \ln(2/\alpha)} \right) \right)^2 \sigma_j(V).$$

**Claim 20.** *For any matrix $H \succeq \rho I \succ 0$, vector $v$, and constant $c \ge 0$ satisfying $H - cI \succ 0$,*

$$\|v\|_{(H-cI)^{-1}} \le \|v\|_{H^{-1}} \sqrt{\frac{\rho}{\rho - c}}.$$

*Proof.* Since $0 \prec \rho I \preceq H$, an application of Claim 16 gives

$$cI \preceq (c/\rho)H \qquad\qquad \text{multiplying by } c/\rho \geq 0$$

$$H - cI \succeq H - (c/\rho)H = \left(\frac{\rho - c}{\rho}\right)H$$

$$(H - cI)^{-1} \preceq \left(\frac{\rho}{\rho - c}\right)H^{-1}.$$

The result follows from the definition of $\|v\|_{H^{-1}}$. $\qquad\qquad\qquad\qquad\qquad\qquad\square$

## B  Discussion and Proofs from Section 3

The proofs in this section are based on those of Abbasi-Yadkori et al. [1], who analyze the LinUCB algorithm with a constant regularizer. The main difference from that work is that, in our case, quantities involving the regularizer must be bounded above or below (as appropriate) by the constants $\rho_{\max}$ and $\rho_{\min}$, respectively. We will make extensive use of Claim 16, which shows that for any two matrices $0 \preceq V \preceq U$, we have $\det V \leq \det U$ and $V^{-1} \succeq U^{-1}$. We start by proving the following proposition about the sizes of the confidence ellipsoids, which illustrates this general idea.

**Proposition 4** (Calculating $\beta_t$). *Suppose Assumptions 3 to 5 hold and let $\rho_{\min}$, $\rho_{\max}$, and $\gamma$ be $(\alpha/2n)$-accurate for some $\alpha \in (0, 1)$ and horizon $n$. Then $(\beta_t)_{t=1}^n$ is $(\alpha, n)$-accurate where*

$$\beta_t := \sigma\sqrt{2\log(2/\alpha) + \log(\det V_t) - d\log(\rho_{\min})} + S\sqrt{\rho_{\max}} + \gamma$$

$$\leq \sigma\sqrt{2\log(2/\alpha) + d\log\left(\frac{\rho_{\max}}{\rho_{\min}} + \frac{tL^2}{d\rho_{\min}}\right)} + S\sqrt{\rho_{\max}} + \gamma. \qquad \textit{(if Assumption 1 also holds)}$$

*Proof.* By definition, $\tilde{\theta}_t = V_t^{-1}\tilde{u}_t$, $\tilde{u}_t = u_t + h_t$, and $u_t = X_{<t}^\mathsf{T}y_{<t}$, so that

$$\begin{aligned}
\theta^* - \tilde{\theta}_t &= \theta^* - V_t^{-1}(X_{<t}^\mathsf{T}y_{<t} + h_t) \\
&= \theta^* - V_t^{-1}(X_{<t}^\mathsf{T}X_{<t}\theta^* + X_{<t}^\mathsf{T}\eta_{t-1} + h_t) && \text{since } y_{<t} = X_{<t}\theta^* + \eta_{t-1} \\
&= \theta^* - V_t^{-1}(V_t\theta^* - H_t\theta^* + z_t + h_t) && \text{defining } z_t := X_{<t}^\mathsf{T}\eta_{t-1} \\
&= V_t^{-1}(H_t\theta^* - z_t - h_t)
\end{aligned}$$

Multiplying both sides by $V_t^{1/2}$ gives

$$\begin{aligned}
V_t^{1/2}(\theta^* - \tilde{\theta}_t) &= V_t^{-1/2}(H_t\theta^* - z_t - h_t) \\
\|\theta^* - \tilde{\theta}_t\|_{V_t} &= \|H_t\theta^* - z_t - h_t\|_{V_t^{-1}} && \text{applying } \|\cdot\| \text{ to both sides} \\
&\leq \|H_t\theta^*\|_{V_t^{-1}} + \|z_t\|_{V_t^{-1}} + \|h_t\|_{V_t^{-1}} && \text{triangle inequality} \\
&\leq \|z_t\|_{V_t^{-1}} + \|H_t\theta^*\|_{H_t^{-1}} + \|h_t\|_{H_t^{-1}} && \text{by Claim 16 since } V_t \succeq H_t \\
&= \|z_t\|_{(G_t + \rho_{\min}I)^{-1}} + \|\theta^*\|_{H_t} + \|h_t\|_{H_t^{-1}}, && \text{since } V_t \succeq G_t + \rho_{\min}I.
\end{aligned}$$

We use a union bound over all $n$ rounds to bound $\|\theta^*\|_{H_t} \leq \sqrt{\|H_t\|}\|\theta^*\| \leq S\sqrt{\rho_{\max}}$ and $\|h_t\|_{H_t^{-1}} \leq \gamma$ with probability at least $1 - \alpha/2$. Finally, by the "self-normalized bound for vector-valued martingales" of Abbasi-Yadkori et al. [1, Theorem 1], with probability $1 - \alpha/2$ for all rounds simultaneously

$$\|z_t\|_{(G_t + \rho_{\min}I)^{-1}} \leq \sigma\sqrt{2\log\frac{2}{\alpha} + \log\frac{\det(G_t + \rho_{\min}I)}{\det \rho_{\min}I}} \leq \sigma\sqrt{2\log\frac{2}{\alpha} + \log\det V_t - d\log\rho_{\min}}.$$

It only remains to show the upper-bound on each $\beta_t$. By Claim 16, we have $\det V_t = \det(G_t + H_t) \leq \det(G_t + \rho_{\max}I)$ and

$$\log\det V_t \leq \log\det(G_t + \rho_{\max}I) \leq d\log(\rho_{\max} + tL^2/d).$$

using the trace-determinant inequality as in the proof of Lemma 22. All the $\beta_t$ are therefore bounded by the constants

$$\bar{\beta}_t := \sigma\sqrt{2\log\frac{2}{\alpha} + d\log\left(\frac{\rho_{\max}}{\rho_{\min}} + \frac{tL^2}{d\rho_{\min}}\right)} + S\sqrt{\rho_{\max}} + \gamma. \qquad\qquad\qquad \square$$

We now take our first steps towards a regret bound by giving a "generic" version that depends only on LinUCB taking "optimistic" actions, the sizes of the confidence sets, and the rewards being bounded. We rely upon the upper bound for each $\beta_t$ shown in the previous proposition. We use the Cauchy-Schwarz inequality to bound the sum of per-round regrets $r_t$ by $\sum r_t^2$; this results in the leading $O(\sqrt{n})$ factor in the regret bound. Our gap-dependent analysis later avoids this, but has other trade-offs.

**Lemma 21** (Generic LinUCB Regret). *Suppose Assumptions 2 and 4 hold (i.e. $|\langle \theta^*, x \rangle| \leq B$ and all $H_t \geq 0$) and $\bar{\beta}_n \geq \max\{\beta_1, \ldots, \beta_n, 1\}$; also assume that $B = 1$. If all the confidence sets $\mathcal{E}_t$ contain $\theta^*$ (i.e., $\|\theta^* - \bar{\theta}_t\|_{V_t} \leq \beta_t$), then the pseudo-regret of Algorithm 1 is bounded by*

$$\widehat{R}_n \leq \bar{\beta}_n \sqrt{4n \sum_{t=1}^{n} \min\{1, \|x_t\|_{V_t^{-1}}^2\}}.$$

*Remark* 2 (On the quantity $B$ appearing in Assumption 2). For the following proofs, we assume (as in this lemma) that Assumption 2 holds with $B = 1$. Eventually, however, our regret bounds end up with a factor $B$; we now explain how. First note that $B$ is trivially at most $LS$ by Cauchy-Schwarz: $|\langle \theta^*, x_t \rangle| \leq \|\theta^*\|\|x_t\| \leq LS$. The case where $B < 1$ yet is some constant is trivial: clearly we can take $B = 1$ without violating the assumption. The case where $B = o(1)$ is actually quite intricate and somewhat "unnatural": while a-priori we know the mean-reward can be as large as $LS$, it is in fact *much* smaller. This means we have to scale down actions, and shrink the entire problem by a sub-constant; and as a result the noise $\sigma$ is actually now *far* larger (it is like $\sigma/B$ in the original setting). While this can be a mere technicality in general, since our leading application is privacy this also means that the bounds on the actual reward we use in Section 4 needs to be scaled by a very large factor. Thus, allowing for ridiculously small $B$ turns into an unnecessary nuisance, and we simply assume that our upper-bound $B$ is not tiny — namely, we assume $B \geq 1$.

It remains to deal with the situation where $B > 1$. In this case, we can pre-process the rewards to the algorithm, scaling them down by a factor of $B$. If we also scale down all the actions $x \in \mathcal{D}_t$, then the rest of the assumptions remain inviolate and the regret bound for $B = 1$ applies. However, this bounds the regret in the scaled problem: in the original problem the rewards and hence regret must be scaled up by a factor of $B$. Note that by only scaling the $\mathcal{D}_t$, we are not modifying any quantities used in the actual algorithm, just the regret bound; this would not be true if we scaled $\theta^*$ (whose maximum norm appears in the $\beta_t$), which is the other possibility to maintain the linearity of rewards.

Indeed, in the scaled-down problem the regret is somewhat lower than the bound because both $L$ and $\sigma$ can be scaled down by $B$ (the noise variance scales proportionally to the reward). For simplicity, however, we refrain from replacing $L$ and $\sigma$ in the upper bound with $L/B$ and $\sigma/B$, respectively.

*Proof of Lemma 21.* At every round $t$, Algorithm 1 selects an "optimistic" action $x_t$ satisfying

$$(x_t, \bar{\theta}_t) \in \underset{(x, \theta) \in \mathcal{D}_t \times \mathcal{E}_t}{\arg\max} \langle \theta, x \rangle. \tag{5}$$

Let $x_t^* \in \arg\max_{x \in \mathcal{D}_t} \langle \theta^*, x \rangle$ be an optimal action and $r_t = \langle \theta^*, x_t^* - x_t \rangle$ be the immediate pseudo-regret suffered for round $t$:

$$
\begin{aligned}
r_t &= \langle \theta^*, x_t^* \rangle - \langle \theta^*, x_t \rangle \\
&\leq \langle \bar{\theta}_t, x_t \rangle - \langle \theta^*, x_t \rangle && \text{from (5) since } (x_t^*, \theta^*) \in \mathcal{D}_t \times \mathcal{E}_t \\
&= \langle \bar{\theta}_t - \theta^*, x_t \rangle \\
&= \langle V_t^{1/2}(\bar{\theta}_t - \theta^*), V_t^{-1/2} x_t \rangle && \text{since } V_t \geq H_t \geq 0 \\
&\leq \|\bar{\theta}_t - \theta^*\|_{V_t} \|x_t\|_{V_t^{-1}} && \text{by Cauchy-Schwarz} \\
&\leq \left( \|\bar{\theta}_t - \tilde{\theta}_t\|_{V_t} + \|\theta^* - \tilde{\theta}_t\|_{V_t} \right) \|x_t\|_{V_t^{-1}} && \text{by the triangle inequality} \\
&\leq 2\beta_t \|x_t\|_{V_t^{-1}} && \text{since } \bar{\theta}_t, \theta^* \in \mathcal{E}_t \\
&\leq 2\bar{\beta}_n \|x_t\|_{V_t^{-1}} && \text{since } \bar{\beta}_n \geq \beta_t.
\end{aligned}
$$

From our assumptions that the mean absolute reward is at most 1 and $\bar{\beta}_n \geq 1$, we also get that $r_t \leq 2 \leq 2\bar{\beta}_n$. Putting these together,

$$r_t \leq 2\bar{\beta}_n \min\{1, \|x_t\|_{V_t^{-1}}\} \tag{6}$$

Now we apply the Cauchy-Schwarz inequality, since $\widehat{R}_n = \langle \mathbf{1}_n, \boldsymbol{r}/n \rangle$, where $\mathbf{1}_n$ is the all-ones vector and $\boldsymbol{r}$ is the vector of per-round regrets:

$$\widehat{R}_n^2 = n^2 \Big(\sum_{t=1}^n \frac{r_t}{n}\Big)^2 \le n^2 \sum_{t=1}^n \frac{r_t^2}{n} = n \sum_{t=1}^n r_t^{\,2} \le 4n\bar{\beta}_n^2 \sum_{t=1}^n \min\{1, \|\boldsymbol{x}_t\|_{V_t^{-1}}^2\}.$$

Taking square roots completes the proof. □

The following technical lemma relates the quantity from the previous result to the volume (i.e. determinant) of the $V_n$ matrix. We will see shortly that the $U_t$ are all lower bounds on the $V_t$.

**Lemma 22** (Elliptical Potential). *Let $\boldsymbol{x}_1, \ldots, \boldsymbol{x}_n \in \mathbb{R}^d$ be vectors with each $\|\boldsymbol{x}_t\| \le L$. Given a positive definite matrix $U_1 \in \mathbb{R}^{d \times d}$, define $U_{t+1} := U_t + \boldsymbol{x}_t \boldsymbol{x}_t^\mathsf{T}$ for all $t$. Then*

$$\sum_{t=1}^n \min\{1, \|\boldsymbol{x}_t\|_{U_t^{-1}}^2\} \le 2 \log \frac{\det U_{n+1}}{\det U_1} \le 2d \log \frac{\operatorname{tr} U_1 + nL^2}{d \det^{1/d} U_1}.$$

*Proof.* We use the fact that $\min\{1, u\} \le 2\log(1 + u)$ for any $u \ge 0$:

$$\sum_{t=1}^n \min\{1, \|\boldsymbol{x}_t\|_{U_t^{-1}}^2\} \le 2 \sum_{t=1}^n \log(1 + \|\boldsymbol{x}_t\|_{U_t^{-1}}^2).$$

We will show that this last summation is $2 \log(\det U_{n+1}/\det U_n)$. For all $t$, we have

$$U_{t+1} = U_t + \boldsymbol{x}_t \boldsymbol{x}_t^\mathsf{T} = U_t^{1/2}\big(I + U_t^{-1/2} \boldsymbol{x}_t \boldsymbol{x}_t^\mathsf{T} U_t^{-1/2}\big) U_t^{1/2}$$

$$\det U_{t+1} = \det U_t \det\big(I + U_t^{-1/2} \boldsymbol{x}_t \boldsymbol{x}_t^\mathsf{T} U_t^{-1/2}\big).$$

Consider the eigenvectors of the matrix $I + \boldsymbol{y}\boldsymbol{y}^\mathsf{T}$ for an arbitrary vector $\boldsymbol{y} \in \mathbb{R}^d$. We know that $\boldsymbol{y}$ itself is an eigenvector with eigenvalue $1 + \|\boldsymbol{y}\|^2$:

$$(I + \boldsymbol{y}\boldsymbol{y}^\mathsf{T})\boldsymbol{y} = \boldsymbol{y} + \boldsymbol{y}\langle \boldsymbol{y}, \boldsymbol{y}\rangle = (1 + \|\boldsymbol{y}\|^2)\boldsymbol{y}.$$

Moreover, since $I + \boldsymbol{y}\boldsymbol{y}^\mathsf{T}$ is symmetric, every other eigenvector $\boldsymbol{u}$ is orthogonal to $\boldsymbol{y}$, so that

$$(I + \boldsymbol{y}\boldsymbol{y}^\mathsf{T})\boldsymbol{u} = \boldsymbol{u} + \boldsymbol{u}\langle \boldsymbol{y}, \boldsymbol{u}\rangle = \boldsymbol{u}.$$

Therefore the only eigenvalues of $I + \boldsymbol{y}\boldsymbol{y}^\mathsf{T}$ are $1 + \|\boldsymbol{y}\|^2$ (with eigenvector $\boldsymbol{y}$) and 1. In our case $\boldsymbol{y} = U_t^{-1/2}\boldsymbol{x}_t$ and $\|\boldsymbol{y}\|^2 = \boldsymbol{x}_t^\mathsf{T} U_t^{-1} \boldsymbol{x}_t = \|\boldsymbol{x}_t\|_{U_t^{-1}}^2$, so we get our first inequality:

$$\det U_{n+1} = \det U_1 \prod_{t=1}^n (1 + \|\boldsymbol{x}_t\|_{U_t^{-1}}^2)$$

$$2 \log \frac{\det U_{n+1}}{\det U_1} = 2 \sum_{t=1}^n \log(1 + \|\boldsymbol{x}_t\|_{U_t^{-1}}^2).$$

To get the second inequality, we apply the arithmetic-geometric mean inequality to the eigenvalues $\lambda_i$ of $U_n$:

$$\det U_n = \prod_{i=1}^d \lambda_i \le \Big(\frac{1}{d}\sum_{i=1}^d \lambda_i\Big)^d = ((1/d)\operatorname{tr} U_n)^d \le ((\operatorname{tr} U_1 + nL^2)/d)^d$$

$$2 \log \frac{\det U_n}{\det U_1} \le 2d \log \frac{\operatorname{tr} U_1 + nL^2}{d \det^{1/d} U_1} \qquad\qquad □$$

We are finally in a position to prove the main regret theorem. The proof is straightforward and essentially comes down to plugging in our preceding results.

**Theorem 5** (Regret of Algorithm 1). *Suppose Assumptions 1 to 5 hold and the $\beta_t$ are as given by Proposition 4. Then with probability at least $1 - \alpha$ the pseudo-regret of Algorithm 1 is bounded by*

$$\widehat{R}_n \le B\sqrt{8n}\left[\sigma\Big(2\log(\tfrac{2}{\alpha}) + d\log\Big(\tfrac{\rho_{\max}}{\rho_{\min}} + \tfrac{nL^2}{d\rho_{\min}}\Big)\Big) + (S\sqrt{\rho_{\max}} + \gamma)\sqrt{d\log\Big(1 + \tfrac{nL^2}{d\rho_{\min}}\Big)}\right] \qquad (2)$$

*Proof.* We restrict ourselves to the event that all the confidence ellipsoids contain $\theta^*$ and all $\rho_{\min} I \preceq H_t \preceq \rho_{\max} I$. Proposition 4 assures us that this happens with probability at least $1 - \alpha$, and

furthermore gives us the bound $\beta_t \le \bar{\beta}_n$:

$$\bar{\beta}_n := \sigma\sqrt{2\log\frac{2}{\alpha} + d\log\left(\frac{\rho_{\max}}{\rho_{\min}} + \frac{nL^2}{d\rho_{\min}}\right)} + S\sqrt{\rho_{\max}} + \gamma.$$

Next, we have $\|x_t\|_{V_t^{-1}} \le \|x_t\|_{(G_t + \rho_{\min}I)^{-1}}$, which applied to the result of Lemma 21 gives, using Lemma 22

$$\widehat{R}_n \le \bar{\beta}_n\sqrt{8dn\log\left(1 + \frac{nL^2}{d\rho_{\min}}\right)}$$

$$\le \sqrt{8n}\left[\sigma\left(2\log\frac{2}{\alpha} + d\log\left(\frac{\rho_{\max}}{\rho_{\min}} + \frac{nL^2}{d\rho_{\min}}\right)\right) + (S\sqrt{\rho_{\max}} + \gamma)\sqrt{d\log\left(1 + \frac{nL^2}{d\rho_{\min}}\right)}\right].$$

The argument outlined in Remark 2 preceding the proof of Lemma 21 tells us how to reintroduce the missing factor of $B$ in this regret bound. □

The proof of the gap-dependent regret bound diverges from the previous proof in only one major way: the gap is used to bound each $r_t$ by $r_t^2/\Delta$. Then the sum of $r_t^2$ is bounded as before; this avoids the $\sqrt{n}$ factor introduced by the use of the Cauchy-Schwarz inequality.

**Theorem 6** (Gap-Dependent Regret of Algorithm 1). *Suppose Assumptions 1 to 5 hold and the $\beta_t$ are as given by Proposition 4. If the optimal actions in every decision set $\mathcal{D}_t$ are separated from the sub-optimal actions by a reward gap of at least $\Delta$, then with probability at least $1 - \alpha$ the pseudo-regret of Algorithm 1 satisfies*

$$\widehat{R}_n \le \frac{8B}{\Delta}\left[\sigma\left(2\log(\tfrac{2}{\alpha}) + d\log\left(\tfrac{\rho_{\max}}{\rho_{\min}} + \tfrac{nL^2}{d\rho_{\min}}\right)\right) + (S\sqrt{\rho_{\max}} + \gamma)\sqrt{d\log\left(1 + \tfrac{nL^2}{d\rho_{\min}}\right)}\right]^2 \qquad (3)$$

*Proof.* Because of the gap assumption, for every round $t$ if the per-round pseudo-regret $r_t \ne 0$ then $r_t \ge \Delta$. We use this fact to decompose the regret in a different way than we did in Lemma 21. The rest of the proof is similar to that of Theorem 5. As before, see Remark 2 preceding the proof of Lemma 21 to introduce the missing $B$ factor.

$$\widehat{R}_n = \sum_{t \in B_n} r_t \le \sum_{t \in B_n} \frac{r_t^2}{\Delta}$$

$$\le \frac{4}{\Delta}\bar{\beta}_n^2 \sum_{t \in B_n} \min\{1, \|x_t\|_{V_t^{-1}}^2\} \qquad\qquad \text{from (6)}$$

$$\le \frac{8}{\Delta}\bar{\beta}_n^2 d\log\left(1 + \frac{nL^2}{d\rho_{\min}}\right)$$

$$\le \frac{8}{\Delta}\left[\sigma\left(2\log\frac{2}{\alpha} + d\log\left(\frac{\rho_{\max}}{\rho_{\min}} + \frac{nL^2}{d\rho_{\min}}\right)\right) + (S\sqrt{\rho_{\max}} + \gamma)\sqrt{d\log\left(1 + \frac{nL^2}{d\rho_{\min}}\right)}\right]^2 \qquad □$$

### B.1 Regret Bounds Open Problem

The first conclusion from these regret bounds is that allowing changing regularizers does not incur significant additional regret, as long as they are bounded both above and below. Broadly speaking, these bounds for contextual linear bandits match those for standard MAB algorithms in terms of their dependence on $n$ and $\Delta$ — just like with UCB, for example, the minimax bound is $O(\sqrt{n})$ and the gap-dependent bound is $O(\log(n)/\Delta)$. However, the dependence on $d$ (which corresponds to the number of arms for the MAB) is much worse, with $O(d)$ in the minimax case and $O(d^2)$ in the gap-dependent case.

It is an interesting open question whether the $O(d^2)$ dependence on $d$ is necessary to achieve $O(\log n)$ gap-dependent regret bounds. As we were unable to prove a lower bound of $\Omega(d^2)$, we resorted to empirically checking the performance of the (non-private) LinUCB on such $\Delta$-gap instances; the results can be found in Section D.1.

## C  Privacy Proofs

We now provide the missing privacy proofs from the main body of the paper. First, we give the omitted proof from Section 4.1.

**Proposition 9.** *Fix any $\alpha > 0$. If for each $t$ the $\boldsymbol{H}_t$ and $\boldsymbol{h}_t$ are generated by the tree-based algorithm with Wishart noise $\mathcal{W}_{d+1}(\tilde{L}^2 \boldsymbol{I}, k)$, then the following are $(\alpha/2n)$-accurate bounds:*

$$\rho_{\min} = \tilde{L}^2 \big(\sqrt{mk} - \sqrt{d} - \sqrt{2\ln(8n/\alpha)}\big)^2,$$
$$\rho_{\max} = \tilde{L}^2 \big(\sqrt{mk} + \sqrt{d} + \sqrt{2\ln(8n/\alpha)}\big)^2,$$
$$\gamma = \tilde{L}\big(\sqrt{d} + \sqrt{2\ln(2n/\alpha)}\big).$$

*Moreover, if we use the shifted regularizer $\boldsymbol{H}_t' := \boldsymbol{H}_t - c\boldsymbol{I}$ with $c$ as given in Eq. (4), then the following are $(\alpha/2n)$-accurate bounds:*

$$\rho_{\min}' = 4\tilde{L}^2 \sqrt{mk}\big(\sqrt{d} + \sqrt{2\ln(8n/\alpha)}\big),$$
$$\rho_{\max}' = 8\tilde{L}^2 \sqrt{mk}\big(\sqrt{d} + \sqrt{2\ln(8n/\alpha)}\big),$$
$$\gamma' = \tilde{L}\sqrt{\sqrt{mk}\big(\sqrt{d} + \sqrt{2\ln(2n/\alpha)}\big)}.$$

*Proof.* Seeing as $\boldsymbol{H}_t \sim \mathcal{W}_d(\tilde{L}^2\boldsymbol{I}, mk)$, straight-forward bounds on the eigenvalues of the Wishart distribution (e.g. [30], Lemma A.3) give that w.p. $\geq 1 - \alpha/2n$ all of the eigenvalues of $\boldsymbol{H}_t$ lie in the interval $\tilde{L}^2\big(\sqrt{mk} \pm \big(\sqrt{d} + \sqrt{2\ln(8n/\alpha)}\big)\big)^2$. To bound $\|\boldsymbol{h}_t\|_{\boldsymbol{H}_t^{-1}}$ we draw back to the definition of the Wishart distribution as the Gram matrix of samples from a multivariate Gaussian $\mathcal{N}(\boldsymbol{0}, \tilde{L}^2\boldsymbol{I})$. Denote this matrix of Gaussians as $[\boldsymbol{Z}; \boldsymbol{z}]$ where $\boldsymbol{Z} \in \mathbb{R}^{mk \times d}$ and $\boldsymbol{z} \in \mathbb{R}^{mk}$, and we have that $\boldsymbol{H}_t = \boldsymbol{Z}^\intercal \boldsymbol{Z}$ and $\boldsymbol{h}_t = \boldsymbol{Z}^\intercal \boldsymbol{z}$, thus $\|\boldsymbol{h}_t\|_{\boldsymbol{H}_t^{-1}} = \sqrt{\boldsymbol{z}^\intercal \boldsymbol{Z}(\boldsymbol{Z}^\intercal \boldsymbol{Z})^{-1}\boldsymbol{Z}^\intercal \boldsymbol{z}}$. The matrix $\boldsymbol{Z}(\boldsymbol{Z}^\intercal \boldsymbol{Z})^{-1}\boldsymbol{Z}^\intercal$ is a projection matrix onto a $d$-dimensional space, and projecting the spherical Gaussian $\boldsymbol{z}$ onto this subspace results in a $d$-dimensional spherical Gaussian. Using concentration bounds on the $\chi^2$-distribution (Claim 17) we have that w.p. $\geq 1 - \alpha/2n$ it holds that $\|\boldsymbol{h}_t\|_{\boldsymbol{H}_t^{-1}} \leq \gamma := \tilde{L}\big(\sqrt{d} + \sqrt{2\ln(2n/\alpha)}\big)$.

It is straightforward to modify these bounds for the shifted regularizer matrix $\boldsymbol{H}_t' := \boldsymbol{H}_t - c\boldsymbol{I}$; the minimum and maximum eigenvalues are bounded as $\rho_{\min}' = \rho_{\min} - c$ and $\rho_{\max}' = \rho_{\max} - c$, respectively. The value of $c$ in Eq. (4) is chosen so that $\rho_{\min}' = \rho_{\min} - c = \rho_{\max} - \rho_{\min} = 4\tilde{L}^2\sqrt{mk}(\sqrt{d} + \sqrt{2\ln(8n/\alpha)})$. It follows that $\rho_{\max}' = \rho_{\max} - c = \rho_{\min}' + \rho_{\max} - \rho_{\min} = 2\rho_{\min}'$. Finally, Claim 20 gives

$$\|\boldsymbol{h}_t\|_{\boldsymbol{H}_t'^{-1}} \leq \|\boldsymbol{h}_t\|_{\boldsymbol{H}_t^{-1}} \sqrt{\rho_{\min}/\rho_{\min}'} \leq \gamma\sqrt{\rho_{\min}/\rho_{\min}'}$$

$$= \tilde{L}\big(\sqrt{d} + \sqrt{2\ln(2n/\alpha)}\big)\sqrt{\frac{\tilde{L}^2\big(\sqrt{mk} - \sqrt{d} - \sqrt{2\ln(8n/\alpha)}\big)^2}{4\tilde{L}^2\sqrt{mk}\big(\sqrt{d} + \sqrt{2\ln(8n/\alpha)}\big)}}$$

$$\leq \tilde{L}\frac{\big(\sqrt{mk} - \sqrt{d} - \sqrt{2\ln(8n/\alpha)}\big)\sqrt{\sqrt{d} + \sqrt{2\ln(2n/\alpha)}}}{4(mk)^{1/4}}$$

$$\leq \tilde{L}\sqrt{\sqrt{mk}\big(\sqrt{d} + \sqrt{2\ln(2n/\alpha)}\big)} =: \gamma' \qquad \square$$

**Theorem 23** (30, Theorem 4.1). *Fix $\varepsilon \in (0, 1)$ and $\delta \in (0, 1/e)$. Let $A \in \mathbb{R}^{n \times p}$ be a matrix whose rows have $l_2$-norm bounded by $\tilde{L}$. Let $W$ be a matrix sampled from the $d$-dimensional Wishart distribution with $k$ degrees of freedom using the scale matrix $\tilde{L}^2\boldsymbol{I}p$ (i.e. $W \sim \mathcal{W}_p(\tilde{L}^2\boldsymbol{I}p, k)$) for $k \geq p + \lfloor \frac{14}{\varepsilon^2} \cdot 2\log(4/\delta)\rfloor$. Then outputting $A^\intercal A + N$ is $(\varepsilon, \delta)$-differentially private with respect to changing a single row of $A$.*

We now give the proof of the lower bound of any private algorithm under the standard notion of differential privacy under continual observation, as discussed in Section 5. First, of course, we need to define this notion. Formally, two sequences $S = \langle(c_1, y_1), \ldots, (c_n, y_n)\rangle$ and $S' = \langle(c_1', y_1'), \ldots, (c_n', y_n')\rangle$ are called neighbors if there exists a single $t$ such that for any $t' \neq t$ we have $(c_{t'}, y_{t'}) = (c_{t'}', y_{t'}')$; and an algorithm $A$ is $(\varepsilon, \delta)$-differentially private if for any two neighboring sequences $S$ and $S'$ and any subsets of sequences of actions $\mathcal{S} \subset \mathcal{A}^n$ it holds that $\mathbb{P}[A(S) \in \mathcal{S}] \leq e^\varepsilon \mathbb{P}[A(S') \in \mathcal{S}] + \delta$. We now prove the following.

**Claim 13.** *For any $\varepsilon < \ln(2)$ and $\delta < 0.25$, any $(\varepsilon, \delta)$-differentially private algorithm A for the contextual bandit problem must incur pseudo-regret of $\Omega(n)$.*

*Proof.* We consider a setting with two arms $\mathcal{A} = \{a^1, a^2\}$ and two possible contexts: $c^1$ which maps $a^1 \mapsto \boldsymbol{\theta}^*$ and $a^2 \mapsto -\boldsymbol{\theta}^*$; and $c^2$ which flips the mapping. Assuming $\|\boldsymbol{\theta}^*\| = 1$ it is evident we incur a pseudo-regret of 2 when pulling arm $a^1$ is under context $c^2$ or pulling arm $a^2$ under $c^1$. Fix a day $t$ and a history of previous inputs and arm pulls $H_{t-1}$. Consider a pair of neighboring sequences that agree on the history $H_{t-1}$ and differ just on day $t$ — in $S$ the context $c_t = c^1$ whereas in $S'$ it is set as $c_t = c^2$. Denote $\mathcal{S}$ as the subset of action sequences that are fixed on the first $t - 1$ days according to $H_{t-1}$, have the $t$-th action be $a^1$ and on days $> t$ may have any action. Thus, applying the guarantee of differential privacy w.r.t to $\mathcal{S}$ we get that $\mathbb{P}[a_t = a^1 \mid S] = \mathbb{P}[A(S) \in \mathcal{S}] \leq e^{\varepsilon} \mathbb{P}[a_t = a^1 \mid S'] + \delta$. Consider an adversary that sets the context of day $t$ to be either $c^1$ or $c^2$ uniformly at random and independently of other days. The pseudo-regret incurred on day $t$ is thus: $2 \cdot \frac{1}{2} \left( \mathbb{P}[a_t = a^2 \mid S] + \mathbb{P}[a_t = a^1 \mid S'] \right) \geq (1 - \mathbb{P}[a_t = a^1 \mid S]) + e^{-\varepsilon}(\mathbb{P}[a_t = a^1 \mid S] - \delta) = 1 + (e^{-\varepsilon} - 1)\mathbb{P}[a_t = a^2 \mid S] - \delta > 1 - 1 \cdot \frac{1}{2} - \frac{1}{4} = \frac{1}{4}$. As the above applies to any day $t$, the algorithm's pseudo-regret is $\geq \frac{n}{4}$ against such random adversary.  □

# D  Experiments

We performed some experiments with synthetic data to characterize the performance of the algorithms in this paper.

**Setting.**  We first describe the common setting used for all the experiments: Given a dimension $d$, we first select $\boldsymbol{\theta}^*$ to be a random unit vector in $\mathbb{R}^d$ (distributed uniformly on the hyper-sphere, so that $S = 1$). Then we construct decision sets of size $K$ ($K = d^2$ in our experiments), consisting of one *optimal action* and $K - 1$ *suboptimal actions*, all of unit length (so $L = 1$). The optimal action is chosen uniformly at random from the $(d-2)$-dimensional set $\{\boldsymbol{x} \in \mathbb{R}^d \mid \|\boldsymbol{x}\| = 1, \langle \boldsymbol{x}, \boldsymbol{\theta}^* \rangle = 0.75\}$. The suboptimal actions are chosen independently and uniformly at random from the $(d - 1)$-dimensional set $\{\boldsymbol{x} \in \mathbb{R}^d \mid \|\boldsymbol{x}\| = 1, \langle \boldsymbol{x}, \boldsymbol{\theta}^* \rangle \in [-0.75, 0.65]\}$. This results in a suboptimality gap of $\Delta = 0.1$, since the optimal arm has mean reward 0.75 and the suboptimal arms have mean rewards in the $[-0.75, 0.65]$ interval. To simulate the contextual bandit setting, a new decision set is sampled before each round.

Figure 1: An example decision set in $\mathbb{R}^3$ with $K = 1000$ actions.

The rewards are either $-1$ or $+1$, with probabilities chosen so that $\mathbb{E}[y_t] = \langle \boldsymbol{x}_t, \boldsymbol{\theta}^* \rangle$. Therefore $\tilde{B} = 1$ and, being bounded in the $[-1, 1]$ interval, the reward distribution is subgaussian with $\sigma^2 = 1$.

The experiments below measure the expected regret in each case; the confidence parameter is $\alpha = 1/n$, which is the usual choice when one wishes to minimize expected regret.

Figure 2: Experiment 1 — Regret over time for varying dimensions.

Figure 3: Experiment 1 — Regret vs. dimension with log–log axes and best-fit line.

## D.1 The Dependency of the Pseudo-Regret on the Dimension for Gap Instances

The first experiment was aimed at the open question of Section B.1, namely whether the gap-dependent regret is $\Omega(d^2)$ in the dimension of the problem. Thus privacy wasn't a concern in this particular setting; rather, our goal was to determine the performance of our general recipe algorithm in a contextual setting with a clear-cut gap. We measured the pseudo-regret of the non-private LinUCB algorithm as a function of the dimension over $n = 10^5$ rounds with the regularizer $\rho = \boldsymbol{I}_{d \times d}$ and $K = d^2$ arms. The values of $d$ were logarithmically spaced in the interval $[4, 64]$. The results of the experiment are plotted in Fig. 2. The two sub-experiments differ only in the reward noise distribution used. In the first, the reward noise is truly a Gaussian with $\sigma^2 = 1$, whereas in the second the reward is $\pm 1$ as described above (subgaussian with $\sigma^2 = 1$). In the latter case, the actual variance in the reward depends on its expectation, and is somewhat lower than 1. This is perhaps why the regret is somewhat lower than with gaussian reward noise.

Figure 3 shows the same results with total accumulated regret plotted against dimension using a log–log scale. The best-fit line on this plot has a slope of roughly 2, clearly pointing to a super-linear dependency on $d$. We conjecture that, in general, the dependency on $d$ is indeed quadratic.

## D.2 Empirical Performance of the Privacy-Preserving Algorithms over Time

This experiment compares the expected regret of the various algorithm variants presented in this paper. The two major privacy-preserving algorithms are based on Wishart noise (Section 4.1) and Gaussian noise (Section 4.2); both were run with privacy parameters $\varepsilon = 1.0$ and $\delta = 0.1$ over a horizon of $n = 5 \times 10^7$ rounds and dimension $d = 5$ and $K = d^2 = 25$. The results are shown in Fig. 4; the curves are truncated after $2 \times 10^7$ rounds because they are essentially flat after this point.

Figure 4: Experiment 2 — Regret over time, with and without forced sub-optimality gaps.

The sub-figures of Fig. 4 show two settings that differ in the sub-optimality gap $\Delta$ between the rewards of the optimal and sub-optimal arms. In the left sub-figure, the algorithms are run in a setting without a structured gap ($\Delta = 0$), where we have not forced all arms to be strictly separated from the optimal arm by a large reward gap. Here, all sub-optimal arms are distributed uniformly on the set $\{x \in \mathbb{R}^d \mid \|x\| = 1, \langle x, \theta^* \rangle \in [-0.75, 0.75]\}$ (and *not* from $[-0.75, 0.65]$ as in the previous experiment). Note that while we cannot guarantee that in *all* rounds there exists a gap between the optimal and sub-optimal arms, it is still true that *in expectation* we should observe a gap of $\Theta(1/\kappa)$ between the optimal arm and the second-best arm (and as $K = 25$ this expected gap is, still, a constant in comparison to $n$). In the right sub-figure, however, the sub-optimal arms are indeed separated by a gap of $\Delta = 0.1$ from the optimal arms; their rewards lie in the interval $[-0.75, 0.65]$ as in the previous experiment. In both cases there is always an optimal arm with reward 0.75.

The figures show the following algorithm variants:

- The `NonPrivate` algorithm is LinUCB with regularizer $\rho = 1.0$. Its regret is too small to be distinguished from the x-axis in this plot.
- The `Gaussian` variant is described in Section 4.2.
- The `Wishart` variant is described in Section 4.1 with the shift given in Eq. (4).
- The `WishartUnshifted` variant is that of Section 4.1 but with no shift.

**Results.** It is apparent that, at least for this setting, the Gaussian noise algorithm outperforms Wishart noise. This shows that while the asymptotic performance of the two algorithms is fairly close, the constants in the Gaussian version of the algorithm are far better than the ones in the Wishart-noise based algorithm.

Furthermore, the performance of the `WishartUnshifted` variant changes significantly between the two cases — it has the worst regret in the no-gap setting ($\Delta = 0$) but, surprisingly, it is statistically indistinguishable from the shifted `Wishart` variant in the large gap instance ($\Delta = 0.1$). We investigate this relationship between the sub-optimality gap and shifted regularizers in the next experiment.

### D.3 Empirical Performance of Shifted Regularizers for Different Suboptimality Gaps

In both the Wishart and Gaussian variants of our algorithm, we use a *shifted* regularization matrix $H_t \pm cI$, choosing the shift parameter $c$ to approximately optimize our regret bound in each case. This optimal shift parameter turns out not to depend on the sub-optimality gap $\Delta$ of the problem instance. The previous experiment showed, however, that in practice the relative performance of the shifted and unshifted Wishart variants changes drastically depending on the gap. In this experiment, we investigate the impact of varying the shift parameter for the Wishart and Gaussian mechanisms under different sub-optimality gaps $\Delta$.

All the parameters are the same as the previous experiment — the only difference is the shift parameter; the results are shown in Fig. 5. The two sub-figures show the performance of the Wishart and Gaussian variants, respectively. The x-axis is a logarithmic scale indicating $\rho_{\min}$, the high-probability lower bound on the minimum eigenvalue of the shifted regularizer matrix. $\rho_{\min}$ serves as a good proxy for the shift parameter because changing one has the effect of shifting the other by the same amount; it

Figure 5: Experiment 3 — Varying shift parameters with different sub-optimality gaps.

has the added benefit of being meaningfully comparable amongst the different algorithm variants. The vertical dotted lines indicate the $\rho_{\min}$ values corresponding to the algorithms from Sections 4.1 and 4.2 for the Wishart and Gaussian variants, respectively; these are also the algorithms examined in the previous experiment. The Gaussian mechanism does not have an unshifted variant.

**Results.** Tuning the shift parameter appears to significantly affect the performance only for problem instances with relatively small or zero sub-optimality gaps. In the large-gap settings, on the other hand, having too much regularization does not seem to increase regret appreciably. The small-gap settings are exactly those in which exploration is crucial, so we conjecture that large regularizers inhibit exploration and thereby incur increased regret.