[Reviews · NeurIPS 2018]

Reviewer 1



Summary of Paper: A contextual linear bandit algorithm which satisfies the jointly differentially private property is proposed. The central idea is to adopt a changing regularizer in order to combine the classic linear-UCB algorithm and the tree based DP algorithm. The proposed algorithm achieves similar regret guarantee as classic linear bandit. Novel lower bounding results are also provided for general private MAB problems. Comments: The paper is well organized and the technical part is generally sound. While the major concern lies on the significance of the studied problem. According to Theorem 5 and 6, the proposed changing regularization approach has a very similar regret guarantee with the classic algorithm. This reflects that employing the jointly DP constraint does not significantly changes the hardness of the original learning problem. The lower bounding results also reflects this suspicion since the the suffered additional regret is also of order O(log(n)/\delta). I feel it’s better to include some discussions of the importance both in theory and practice for this DP contextual linear bandit setting. I also suggests some improvement for the online shopping example discussed in the introduction. In line 39-41, I am quite confused with the actual meaning of “without revealing preference to all users” since to my understanding, the user preference is only revealed to the system instead of all other users under this scenario. It is better to transform the necessity of privacy to the system side. ----- after rebuttal ----- I believe the feedback has addressed my main concerns.

Reviewer 2



This work describes algorithms for differentially private stochastic contextual linear bandits, and analyzes regret bounds for the same. Privacy in this context means protection against inference of both rewards and inference. The paper first notes that from the action taken in T round it is possible to infer the context even in relatively benign instances. Therefore, it proposes a (reasonable) privacy model where the privacy adversary attempting to infer the context in round t is not aware of the decision/action in all rounds but round t. The algorithmic techniques to achieve this are quite straight-forward -- do linUCB on estimates of cumulative sums made private by tree based aggregation protocol. What to like? 1. It identifies a reasonable privacy model -- since the obvious one leads to linear regret. 2. The problem is important in terms of the number of applications that use the paradigm. I'd even say that it is more practical than typical k-arm stochastic/adversarial bandits that have been studied in context of DP so far. 3. I appreciate the note on lower bounds, esp the lower bound on excess regret for the private variant. Such bounds were somehow missing from the literature so far. I'd like to see a couple of comparisons beyond the ones listed. 1. (Neel, Roth) Mitigating Bias in Adaptive Data Gathering via Differential Privacy. This work used DP to control the bais of empirical means of arms in bandit settings. In the process, they also establish private versions of linUCB. Only the rewards are made private in this setting. 2. (Agarwal, Singh) The Price of Differential Privacy For Online Learning. Here the authors establish regret bounds for full-information online linear learning that has the same regret as the non-private setting in the dominant T^0.5 term. I'm wondering if the authors can comment if their excess regret lower bounds can (for example) say that for the experts settings an excess regret of O((number of expers)/\eps) is necessary. - After the authors' comments. - Score unrevised.

Reviewer 3



Summary: In this paper the authors study how to achieve differential privacy in the contextual linear bandits setting. The authors first show that the standard notion of privacy in this setting leads to the linear regret and then they adopt the more relaxed notion of joint differential privacy. They propose the Linear UCB with changing perturbations algorithm as a general framework for achieving this joint differential privacy while achieving a reasonable regret guarantees. The authors provide general regret bounds for their proposed algorithm under some assumptions in section 3. Later in section 4, they explain two different ways for perturbing the estimations in order to achieve joint differential privacy and they combine them with the results in section 3 to provide regret bounds for them. Finally, in section 5 of the paper, the authors provide some lower bounds and show that any algorithm that achieves differential privacy is required to incur an extra regret term. Evaluation: The paper is very well-written and well-motivated. This work draws an interesting connection between the contextual linear bandit literature and the differential privacy literature. In order to achieve algorithms that can provide privacy and low-regret simultaneously, the authors: (1) derive general regret bounds for their proposed algorithm (which can be found in Proposition 4) and (2) show that with reasonable perturbations the privacy can also be achieved. From the technical analysis, (1) is an easy extension of analysis in [1] and for (2) authors use the result in [2] or the concentration bounds on \chi^2 distribution. Therefore, the technical analysis does not seem to be very different that what existed in the literature. On the other hand, the derivation of lower bounds seem to be pretty novel. Also, I like the fact the proposed algorithm is very general and authors show two different ways to make it work. Finally, I want to emphasize the clarity of this work. Gathering all these points together, I recommend this paper to be accepted to the conference. One slight comment: the citations [28] and [29] in the paper are the same. [1] Abbasi-Yadkori, Y., Pál, D., & Szepesvári, C. (2011). Improved algorithms for linear stochastic bandits. In Advances in Neural Information Processing Systems (pp. 2312-2320). [2] Sheffet, O. (2015). Private approximations of the 2nd-moment matrix using existing techniques in linear regression. arXiv preprint arXiv:1507.00056.